# The protective effect of DMI on hippocampus EEG, behavioral and biochemical parameters in hypoxia-induced seizure on neonatal period

Shadi Nazarizadeh[1], Zohreh Ghotbeddin[1,2]*, Samireh Ghafouri[3,4], Alireza Sarkaki[3,4]

1 Department of Basic Sciences, Faculty of Veterinary Medicine, Shahid Chamran University of Ahvaz, Ahvaz, Iran, 2 Stem Cell and Transgenic Technology Research Center, Shahid Chamran University of Ahvaz, Ahvaz, Iran, 3 Department of Physiology, Medicine Faculty, Ahvaz Jundishapur University of Medical Sciences, Ahvaz, Iran, 4 Persian Gulf Physiology Research Center, Basic Medical Sciences Research Institute, Medicinal Plants Research Center, Ahvaz Jundishapur University of Medical Sciences, Ahvaz, Iran

* z.ghotbeddin@scu.ac.ir

**Data Availability Statement:** All relevant data are within the paper and its Supporting Information files.

## Abstract

Hypoxia-Induced Neonatal Seizure (HINS) is a prevalent type of seizure in infants caused by hypoxic conditions, which can lead to an increased risk of epilepsy, learning disabilities, and cognitive impairments later in life. This study focuses on examining the effects of dimethyl itaconate (DMI) on cognition, motor coordination, and anxiety-like behavior in male rats that have experienced HINS. 42 male Wistar newborn rats (PND10) were randomly divided into six groups (n = 7). 1) Control (Vehicle only); received DMI solvent (0.1ml) without applying hypoxia. 2–3) DMI; receiving (20 and 50 mg/kg; i.p). 4) HINS; they were placed in a hypoxia chamber with 7% oxygen and 93% nitrogen concentration for 15 minutes. 5–6) DMI+HINS; received DMI (20 and 50 mg/kg; i.p) 24h before hypoxia. Behavioral tests including; Novel object recognition test, Rotarod, Parallel bar, Open field and elevated plus maze (EPM); started at age 45 after birth. After behavioral tests, the hippocampal CA1 region local EEG was recorded in all groups. Then the brain hippocampus tissue was isolated and the amount of MDA, SOD, NO, and Thiol was measured by ELISA method. Data showed that the administration of DMI improved motor symptoms, anxiety-like behaviors, and cognition in HINS rats (p<0.05). EEG power in the HINS group decreased significantly compared to other experimental groups (p<0.05). Biochemical observations showed that DMI significantly reduced oxidative stress and inflammation in the hippocampal tissue of HINS rats (p<0.05). Increased hippocampal oxidative stress and inflammation can be effective in the occurrence of behavioral disorders observed in HINS rats. While DMI improved these behavioral impairments by reducing oxidative stress and inflammation.

## Introduction

Hypoxia refers to a condition where there is a decrease in the supply of oxygen to the body's tissues. In neonates, hypoxia can occur due to various reasons, such as birth complications or

**Funding:** The author(s) received no specific funding for this work.

**Competing interests:** The authors have declared that no competing interests exist.

respiratory distress. Hypoxia Induced Neonatal Seizures (HINS) are seizures that occur in newborns as a result of oxygen deprivation during or soon after birth [1]. The occurrence of these seizures during infancy can increase the susceptibility to epilepsy during adulthood and also throughout life [1, 2], which is associated with cognitive impairment [3, 4]. People with HINS are resistant to clinically used conventional anticonvulsants, suggesting that mechanisms that may be involved in neonatal seizures are different from those in adults [5]. Since these seizures occur during the neonatal period, which is very sensitive in terms of the development of synapses [3], it is suggested that HINS can change the developmental pattern of synaptic plasticity, which is the basic mechanism of memory and learning [6]. Studies showed that the hippocampus, a brain area that has a key role in cognition, is highly susceptible to damage when faced with hypoxia [7, 8]. Therefore, it seems that the epileptogenic effects of hypoxia are due to permanent consequences on excitability and plasticity in the hippocampal neuronal network [9]. The exact reason behind the negative impact of seizures on learning and memory remains uncertain [10]. Although the etiology of epilepsy is unknown, many evidences suggest that oxidative stress and inflammation in brain play a crucial role in the process of epileptogenesis [11, 12]. Oxidative stress has been shown to be associated with changes in signaling pathways of reactive oxygen species (ROS), reactive nitrogen species (RNS), and nitric oxide (NO) [13].

Hypoxia is a condition characterized by reduced oxygen supply to the body's tissues, which can occur in neonates due to factors like birth complications or respiratory distress. Hypoxia Induced Neonatal Seizures (HINS) are seizures that arise from oxygen deprivation at or shortly after birth. These seizures can increase the risk of developing epilepsy and cognitive impairments later in life. HINS are often resistant to conventional anticonvulsants, suggesting distinct underlying mechanisms compared to adult seizures. The neonatal period is critical for synaptic development, and HINS may alter synaptic plasticity, which is essential for memory and learning. The hippocampus, vital for cognition, is particularly vulnerable to damage from hypoxia. This damage may lead to increased excitability and altered plasticity in hippocampal networks, contributing to the epileptogenic effects of hypoxia. Although the reasons for the negative impact of seizures on learning and memory are not fully understood, oxidative stress and inflammation in the brain are believed to play significant roles in epileptogenesis, with changes in signaling pathways involving reactive oxygen species (ROS), reactive nitrogen species (RNS), and nitric oxide (NO) being implicated.

One contribution has been the use of animal models to understand the seizure mechanisms. In an experiment, an increase in NO synthesis was observed, which resulted in apoptosis and cell death in the hippocampal CA3 area after kainic acid was injected directly into the region [14]. In the central nervous system, oxidative/nitrative stress and neuroinflammation underlie the pathogenesis of neurological diseases, stroke-related dementia, aging, cancer, and epilepsy. The brain is particularly sensitive to free radicals. In addition, the activity of antioxidant enzymes, such as catalase or glutathione peroxidase, is about ten times lower in the brain than in other tissues. As a result, the free radicals produced in the brain may disrupt the function of neurotransmitters [15, 16]. The cell membrane is rich in unsaturated fatty acids and can be the target of lipid peroxidation. Therefore, the membrane fluidity decreases and changes membrane transporters [16].

Oxidative stress and neuroinflammation are common in patients with various types of epilepsy. Low oxidative stress levels are linked to reduced inflammation, while high levels cause the expression of IL-1β and the generation of an inflammatory signal. It has been reported that an inflammatory response occurs right after hypoxia and lasts for days or even weeks [17, 18]. Seizures are associated with the rapid release of inflammatory cytokines, such as IL-6, IL-1β, and TNF-α [19]. According to earlier research, epilepsy is associated with increased expression

of inflammatory cytokines; therefore, reducing the expression of these factors will considerably lessen the severity of epilepsy [20, 21]. Since inflammation caused by oxidative stress increases the probability of epilepsy, treatment with antioxidants (such as inhibitors of ROS-inducing enzymes) may prevent some epileptogenic processes in the brain [22]. In inflammatory conditions, the accumulation of itaconate, a tricarboxylic acid product, occurs in large quantities [23]. Itaconate is described as a metabolite that negatively regulates oxidative stress and inflammation [24] and can reduce the production of ROS and inflammatory cytokines [25, 26]. One of the cell-permeable derivatives of itaconate is dimethyl itaconate (DMI) [27, 28]. This compound selectively inhibits inflammatory cytokines such as IL-6 and IL-2 [29].

The high prevalence of HINS in human societies and the fact that these patients show resistance to anticonvulsant drugs used in adults during the postnatal period, it is necessary to provide a treatment strategy for these patients. Following creating the HINS model in male rats, by increasing the antioxidant and anti-inflammatory capacity in the brain via intraperitoneal injection of dimethyl itaconate during infancy, we can prevent damage to nerve cells and behavioral disorders in the maturity of these animals. Therefore, this study aimed to investigate the effect of dimethyl itaconate on cognition, motor coordination, and anxiety-like behavior of male rats affected by neonatal seizures.

## Materials and methods

The NIH guidelines for the care and use of laboratory animals were followed, and all procedures were created and carried out in accordance with the recommendations made by the Ethics Committee at Shahid Chamran University of Ahvaz. 42 male Wistar neonatal rats in total were used. The rats were kept in an environment with free access to food and water, and a temperature of 23±2˚C and a 12-hour light/dark cycle. Based on comparable preliminary research, the rats were randomly assigned to six groups, each containing seven animals.

### Experimental groups

The experimental groups included in this study were as follows: Control, DMI 20, DMI 50, Hypoxia, DMI 20 + Hypoxia, and DMI 50 + Hypoxia. The Control rats received only the DMI solvent without hypoxia induction but were placed in the hypoxia chamber for 15 minutes on postnatal day 10 (P10). Instead of hypoxia, intraperitoneal injections of dimethyl itaconate at doses of 20 and 50 mg/kg were given to the DMI 20 and DMI 50 Groups. Once P10 received an intraperitoneal saline injection, the Hypoxia group was exposed to a gas mixture consisting of 7% oxygen and 93% nitrogen to induce hypoxia. The DMI 20 + Hypoxia and DMI 50 + Hypoxia groups received dimethyl itaconate before exposure to the hypoxia chamber on P10. Seizures were induced using a model proposed by Jensen in 1991 [30].

### Behavioral tests

Behavioral tests, including novel object recognition test, rotarod, parallel bar, open field, and elevated plus maze, were used to assay cognition, motor coordination, balance, motor activity, and anxiety-like behaviors, respectively. Behavioral tests were done in adult rats. The timeline of the experiment is shown in Fig 1.

### Novel Object Recognition Test (NORT)

In the novel object recognition test, a square-shaped open field arena measuring 50×50×30 cm with a camera positioned above to record the animal's behavior was used. The experiment

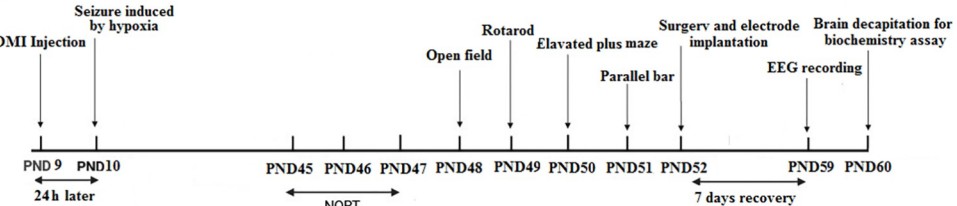

**Fig 1. The timeline of the experiment.**

consisted of three phases: habituation, training, and testing, conducted in a quiet environment with constant lighting. Exploratory behavior was defined as sniffing or touching the novel object within a 2 cm range using the nose or forehead. Each session was recorded on video, and the manual stopwatch was used to measure the exploration time for both the familiar and novel objects. In order to eliminate smell cues, a 10% alcohol solution was used to clean the objects in between trials. Memory evaluation was done using the difference scores, which show how long it took to explore new versus well-known objects. This experiment focused on evaluating the Discrimination index (DI), which was computed in the following way [31]:

Discrimination index (DI) = (time spent exploring the novel object in the third phase / total exploration time in the third phase for both objects) × 100.

## Open Field test (OF)

During the open field test, the animal's movement within a 50 x 50 x 30 cm cube-shaped open field box with a wooden floor was measured and examined for movement patterns. During a five-minute period, the frequency of rearing—standing on the hind legs—and self-grooming behavior were noted. Every experiment was carried out between 9 a.m. and 2 p.m. during the light phase [32].

## Rotarod

Animals were positioned on a rotating horizontal rod with a diameter of 5 cm to evaluate motor coordination and balance using the rotarod apparatus. For 300 seconds, the rod's rotational speed increased gradually from 5 to 45 revolutions per minute. For every animal, the amount of time spent keeping their balance and staying on the rod was noted. Each animal was initially given two chances to become accustomed and adjust to the device. Then, for resting purposes, three more trials were run at a 2-minute interval. They computed the average time on the rod [33].

## Elevated Plus Maze (EPM)

To evaluate the anxiety-like behavior, the elevated plus maze apparatus was used. The apparatus consisted of two open arms and two enclosed arms positioned opposite to each other, with an elevation of 50 cm from the floor. Anxiety-like behavior was defined as the movement into closed arms. The principle of this test relied on the animals' fear of heights and entering the closed arms. Increased exploration and entry into the open arms indicated reduced anxiety. The duration of presence and the number of entries into the open and closed arms were measured as indicators of anxiety and locomotor activity, respectively. To prevent learning, each animal was subjected to the test only once. Following each trial, the apparatus was cleaned by using 10% alcohol [34].

## Parallel bar test

In the parallel bar test, a device consisting of two parallel bars, each 1 meter long with a diameter of 4 mm, placed 30 mm apart and elevated 60 cm above the floor, was used. The rat was positioned in the center of the bars with its head facing downwards, grasping the bars with its forelimbs and hindlimbs. Scoring was based on the time taken for the rat to start rotating from the initial position parallel to the bars until it moved toward one of the device's ends. If the animal remained stationary on the bars for less than 5 seconds and fell, the trial was repeated. The maximum trial duration was set at 120 seconds [35].

## EEG power evaluation

A combination of xylazine and ketamine was used to anesthetize 250–270 g Wistar rats. It was me who anesthetized the animals. Analgesia and sedation are produced by a ketamine and xylazine injection. The animals' tolerance level and weight determine the method's dosage and duration. One way to induce sedation for 30 to 45 minutes is to combine 80–120 mg/kg of ketamine with 5–10 mg/kg of xylazine. Following fur removal on the head, the scalp was grasped above the eyes and pulled backward to the occipital bone, and subsequently incised with scissors. Identification of bregma and lambda points was achieved by applying oxygenated water-soaked cotton to remove the connective tissues on the skull. The skull was then stabilized and positioned using a stereotaxic frame. Precise coordinates (6.2 mm DV, 2.0 mm ML, and 8.3 mm AP) for targeting the hippocampus CA1 region were determined based on a relevant atlas [36]. Using the determined AP and ML coordinates, a precise location above the hippocampus was identified and drilled using a dental drill. A stainless steel electrode with a Teflon insulation coating, measuring 0.005 inches in inner diameter, was guided into the skull at the desired DV depth using a needle and subsequently secured with dental cement. After the cement had dried, the animals were detached from the apparatus and transferred to individual designated cages for further experimentation and observation.

The animals' EEGs were recorded seven days after they had recovered. To acclimate the animals to the surroundings, each animal was given a 15-minute period in the recording chamber. They were subsequently linked to the ML135 Animal Bio Amplifier. The recording was performed with a sampling frequency of 400 Hz, a low-pass filter below 3.0 Hz, and a high-pass filter above 70 Hz. The electrode signal was transferred to the Animal Bio Amplifier and then fed to the 4-channel PowerLab device (AD Instrument Co. Australia). From there, it was transferred to a computer with the Lab Chart 7 software installed for storage and analysis. Finally, the raw EEG voltage for each recording was calculated by taking three 5-second epochs, and the power of different brain waves was computed and compared among different groups.

## Oxidative stress, inflammation, and biochemical parameters

Thirty minutes after EEG recording, the rats were anesthetized with $CO_2$ gas. Following complete anesthesia, the skull was opened, and the brain was extracted. The right and left hippocampi were isolated, stored in microtubes, and immediately transferred to a -70°C freezer for subsequent experiments.

The levels of total thiol groups and the hippocampal concentration of malondialdehyde (MDA), a marker of lipid peroxidation, were measured. The absorption at a wavelength of 535 nm that resulted from the reaction of MDA and thiobarbituric acid (TBA) was measured [37]. Additionally, to evaluate thiol groups, 5'-Dithiobis (2-nitrobenzoic acid) (DTNB), a reagent that reacts with the SH group, was used. The absorption was measured at 412 nm. Therefore, millimoles (mM) were used to express the MDA concentrations and total thiol levels [38].

Calorimetric assessment was used to determine the amount of superoxide dismutase (SOD) present within cells. A unit of SOD was defined as the quantity of enzyme required to impede the MTT reduction rate by fifty percent. Units per milligram of protein were used to calculate the SOD activity, which was measured at 570 nm wavelength [39]. Utilizing the ELISA method in accordance with the manufacturer's instructions, the nitric oxide (NO) concentration was measured. (Anasell, Iran). The nitric oxide concentration was computed in accordance with the manufacturer's instructions after the optical density (OD) was measured at 570 nm wavelength. Plotting the calibration curve and standard concentration values in Excel was followed by normalizing the concentration to umol/ml. NO was measured within the range of 0 to 2000 umol/ml [40].

## Results

### Novel object recognition test

The control group had a significantly higher discrimination index (DI) than the hypoxia, DMI20+ hypoxia ($p < 0.001$), DMI20 group ($p < 0.01$), DMI50 ($p < 0.05$), and DMI50+ hypoxia groups ($p < 0.01$) groups (Fig 2A). In the hypoxia groups, the mean difference score was

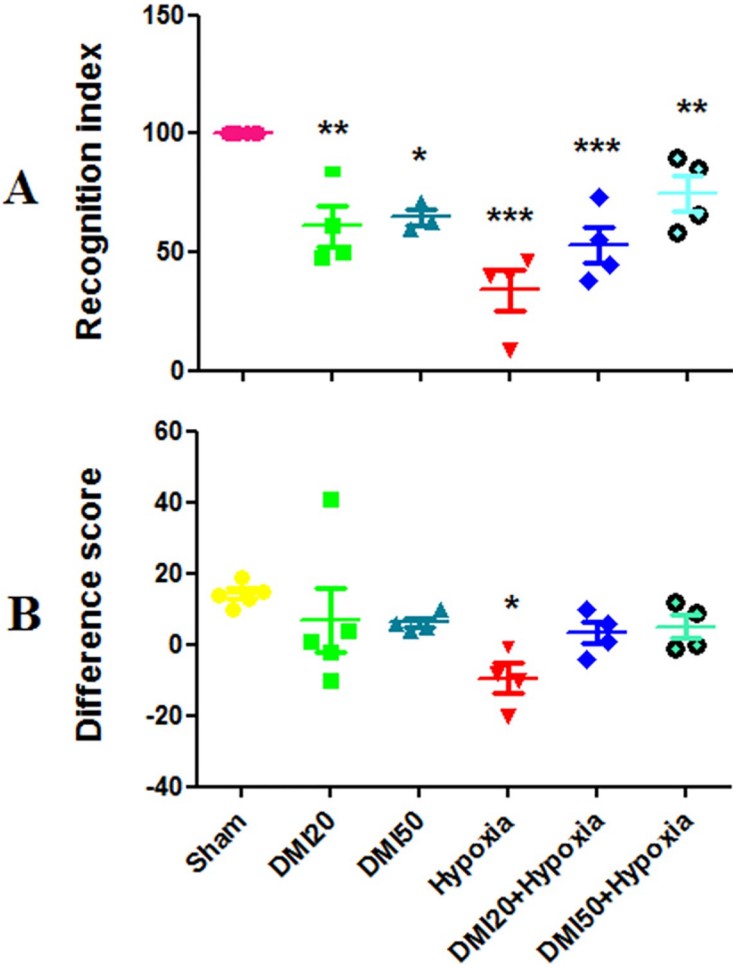

**Fig 2.** A comparison between the experimental groups' discrimination index (percent) (A) and mean difference score (s) (B). *, **, and *** respectively indicate significant differences at the level of ($p < 0.05$), ($p < 0.01$), and ($p < 0.001$) compared to the control group. Data are presented as the mean ± SEM (n = 7).

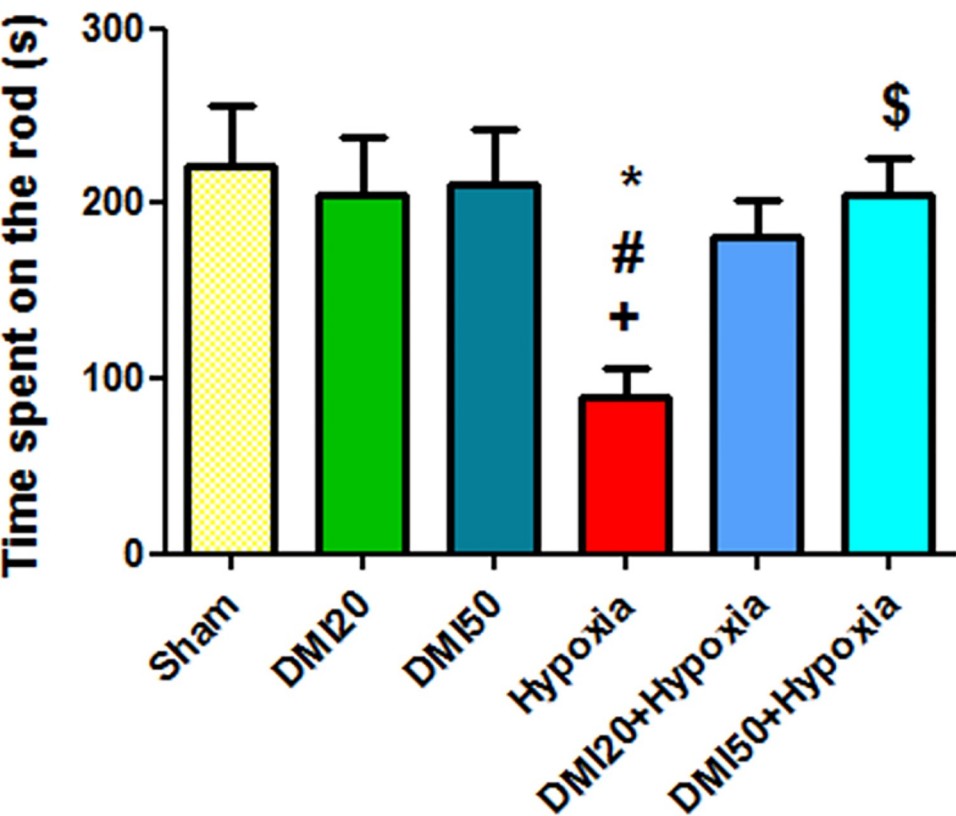

**Fig 3. Comparing the average amount of time (in seconds) that each experimental group spent using the rotarod.**
When compared to the control, DMI50, and DMI20 groups, respectively, *, +, and ++ exhibit significant differences at
the level of (p<0.05). A significant difference between the DMI50+hypoxia and the hypoxia group is indicated by the $
symbol. The mean ± SEM (n = 7) is used to present the data.

approaching negative values and was lower than zero. On the other hand, it increased to positive values in the other groups. In comparison to the control group, the hypoxia group spent less time near the unfamiliar object and more time near the familiar object (p<0.05) (Fig 2B).

## Motor coordination

As has been shown in Fig 3, the average of balance maintenance and motor coordination on the rotating rod in the hypoxia group were significantly lower than the control, DMI20, and DMI50 groups (p<0.05), but in the motor coordination in the DMI50+hypoxia was significantly increased compared to the hypoxia group (p<0.05) (Fig 3). The parallel bars test results showed that the hypoxia group needed the least amount of time to maintain equilibrium. It indicates that, while this difference was not statistically significant, the hypoxia group fell from the parallel bar earlier than the other groups (Fig 4).

## Anxiety-like behavior

Elevated plus maze and open-field tests were used to measure the anxiety-like behavior. Comparing the hypoxia group to the control (p<0.01) and DMI20 (p<0.05), the number of groomings in the open field test increased significantly. DMI 20 (p<0.01) and DMI50 (p<0.001) for several grooming measures were substantially lower in the hypoxia groups than in the hypoxia group (Fig 5A). The hypoxia group experienced a significantly lower number of unsupported

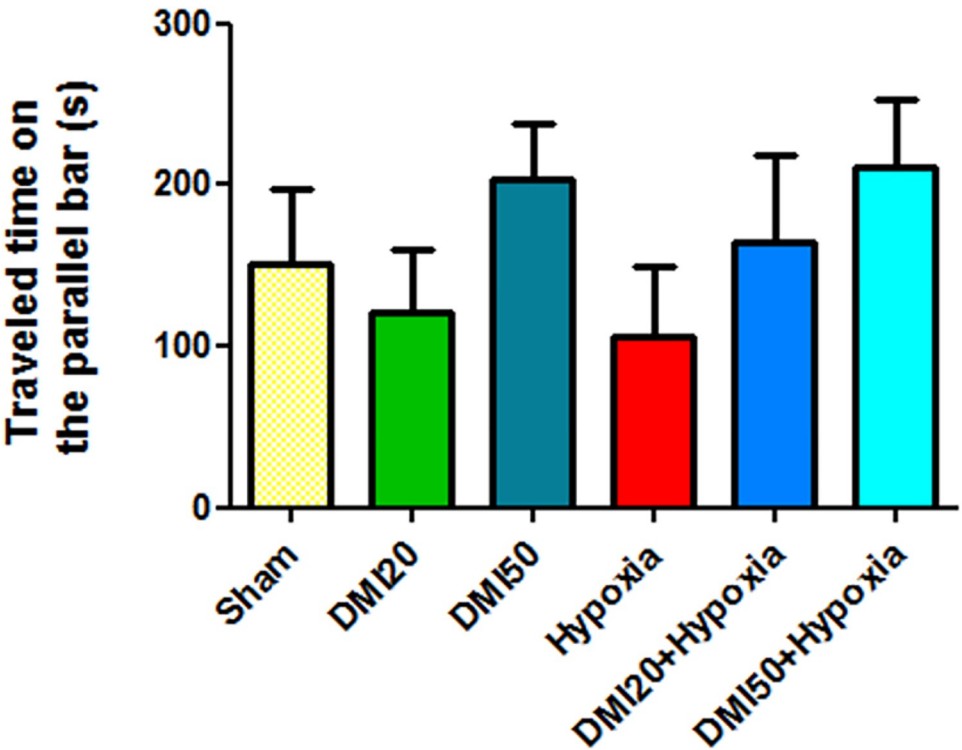

**Fig 4. Compare the mean travel time (in seconds) of the experimental groups on the parallel bar.** *, ++, and + exhibit significant differences from the control, DMI50, and DMI20 groups, respectively, at the level of ($p<0.05$). $ denotes a statistically significant distinction between the hypoxia group and the DMI50+hypoxia group. The data (n = 7) are shown as the mean ± SEM.

rearings ($p<0.01$) compared to the control, DMI20 ($p<0.001$), and DMI50 ($p<0.01$). Additionally, Fig 5B shows that the average amount of unsupported rearing in the DMI50+hypoxia group was lower than that of the control and DMI50 groups ($p<0.05$). This was observed in comparison to the DMI20 group ($p<0.01$) and the DMI20+hypoxia as well. The amount of rearing that was supported in each group did not significantly differ from the other (Fig 5C).

## Oxidative stress and inflammation

The amounts of MDA, SOD, NO, and total thiol in the hippocampus tissue were measured in order to assess oxidative stress and inflammation. The hypoxia group had a significantly higher mean MDA concentration than the control group ($p<0.05$, Fig 7A). Compared to the DMI20 group, the hypoxia group's SOD activity was significantly lower ($p<0.05$, Fig 7B) The hypoxia group's NO concentration was considerably higher than that of the DMI20 and control groups ($p<0.05$, Fig 7C) in addition, the hypoxia group's total thiol level was significantly lower than the control group's ($p<0.05$, Fig 7D).

## Electroencephalography (EEG)

Gamma, beta, alpha, theta, and delta wave powers as well as the raw EEG power were measured and compared between the groups. In comparison to the control ($p<0.001$), DMI20, and DMI50 ($p<0.01$) groups, the raw EEG power dropped significantly in the hypoxia group. EEG power was significantly increased by pre-treatment with DMI20 ($p<0.05$) and DMI50 ($p<0.01$) in comparison to the hypoxia group (Figs 8A and 9). There was no discernible

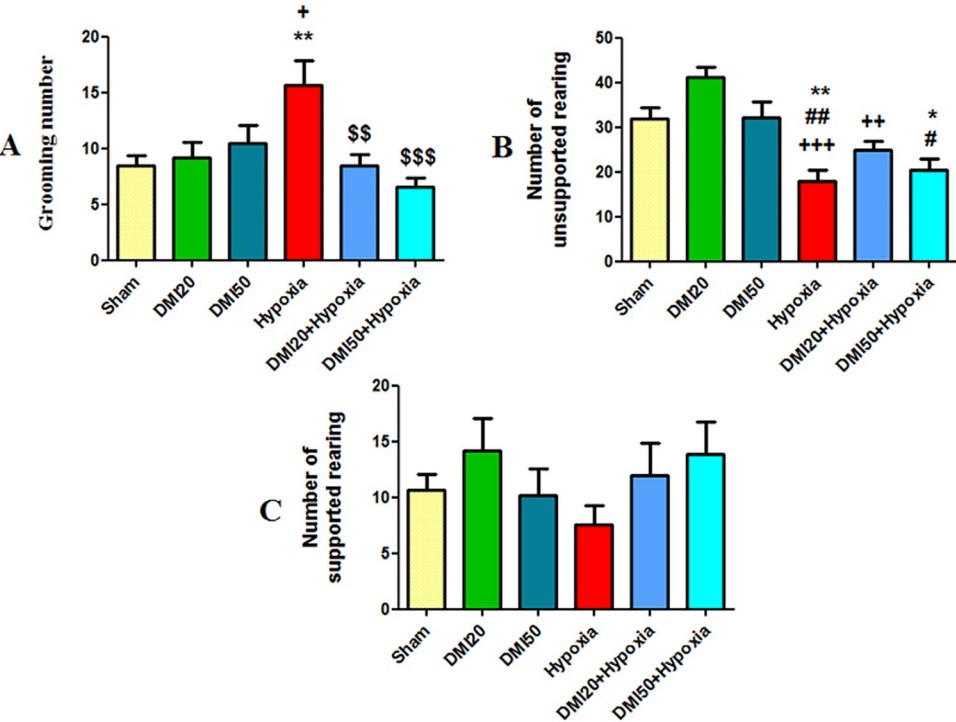

**Fig 5.** The average of grooming number (A), unsupported (B), and supported rearing (C), in the openfield test. When compared to the control group, * and **signify a statistically significant difference at the (p< 0.05) and (p< 0.01) levels. Significant differences from the DMI20 group are indicated by + (p<0.05), ++ (p<0.01), and +++ (p<0.001). There are significant differences (p< 0.01 and p< 0.05) when compared to the DMI50 group. When compared to the hypoxia groups, $ $ and $ $ $ indicate significant differences at the (p<0.01) and (p<0.001) levels, respectively. The data (n = 7) is shown as the mean ± SEM. All groups showed an average of more than 40% of entries into the open arms in the elevated plus maze test; the hypoxia group's entry percentage, on the other hand, was less than 30%. The hypoxia group's average entry into the open arms was notably lower than the control group's (p<0.05, Fig 6A). Additionally, compared to the control group, the hypoxia group showed a significantly lower average percentage of time spent in the open arms (p<0.05, Fig 6B).

variation in gamma wave power between the groups (Fig 8B). The hypoxic groups pretreated with DMI50 showed a significant increase in beta wave power (p<0.05) when compared to the hypoxia group, but the hypoxic groups compared to the control group showed a significant decrease in beta wave power (p<0.05). (Figs 8C and 9). Fig 8D shows that the alpha wave power in the hypoxia group was significantly lower (p<0.05) than in the DMI50 group. Figs 8E and 9 show that the hypoxia group experienced a decrease in theta wave power (p<0.05) when compared to the control group. In the hypoxia group, delta wave power was significantly higher than in the control, DMI20, and DMI50 groups (p<0.05). However, in both hypoxia groups, the pretreatment with DMI20 and DMI50 was significantly lower than in the hypoxia group (p<0.05) (Figs 8F and 9). The EEG power images (Crude local EEG, Beta, Theta and Delta) band traces in the experimental groups.

## Discussion

This study investigated the effects of DMI in newborn rats subjected to hypoxia-induced seizures, on cognitive impairment, motor coordination, anxiety-like behaviors, oxidative stress, and EEG recordings. The hypoxia group showed significantly lower performance in the novel object recognition test, indicating cognitive deficits, which were notably alleviated in DMI-treated hypoxia rats. Prior research supports these findings, linking hypoxia to long-term

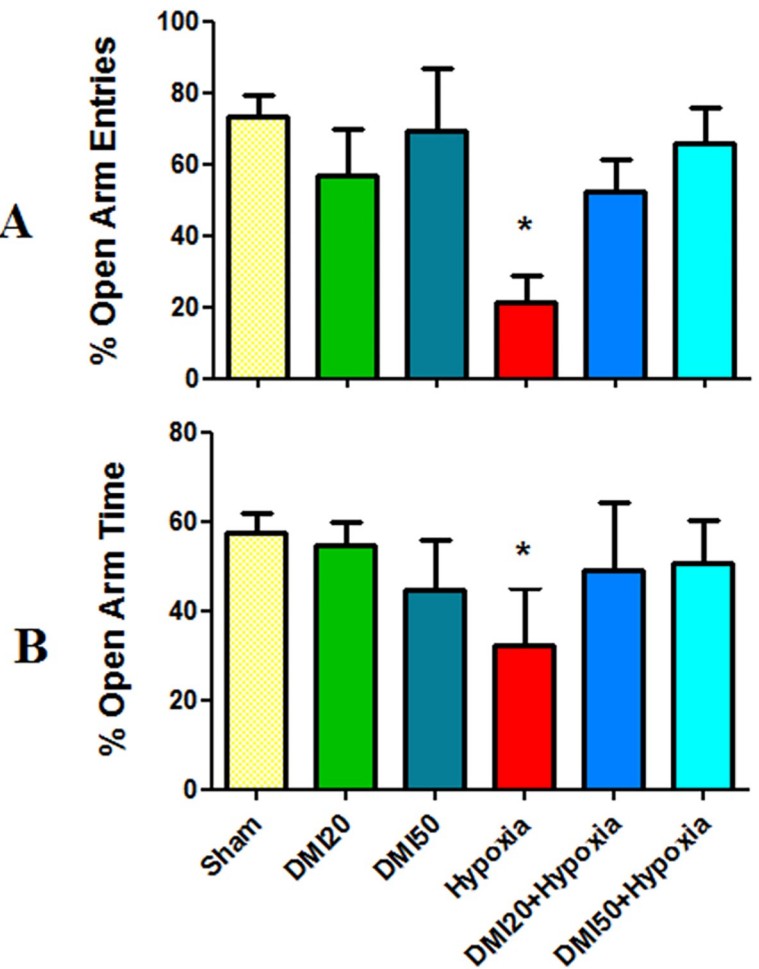

**Fig 6.** Average of open arm entries (A) and time spent in the open arm (B) in the elevated plus maze test. * indicates a statistically significant variation from the control at the level of (p<0.05). The data (n = 7) are shown as the mean ± SEM.

cognitive decline and seizures due to neuronal damage [41, 42]. Additionally, DMI has been shown to enhance cognitive function and synaptic integrity in various models [43].

Motor coordination assessments via the rotarod test revealed that hypoxia significantly impaired balance, but DMI treatment improved balance retention time in hypoxic rats. Overall, DMI demonstrates potential therapeutic benefits in mitigating cognitive and motor deficits associated with hypoxia in neonatal rats.

The evaluation of anxiety-like behaviors in rats exposed to hypoxia was conducted using the open field and elevated plus maze tests, revealing significant increases in anxiety-related behaviors. Treatment with DMI appeared to mitigate these anxiety-like behaviors. These findings align with previous studies by Gailus et al. and Rodriguez-Alvarez et al., which also reported anxiety-related behaviors following hypoxia exposure in animals at various ages and timeframes post-exposure [44].

In another part of this work, to investigate the possible mechanism of hypoxia on behavioral disorders and seizures, oxidative stress parameters such as MDA, SOD, NO, and total thiol were measured. Our results showed that hypoxia increased oxidative stress.

In 2013, Sab et al. also indicated increased oxidative stress, elevated NO production, and disrupted GPx activity in adult mice whose mothers were exposed to hypoxia during

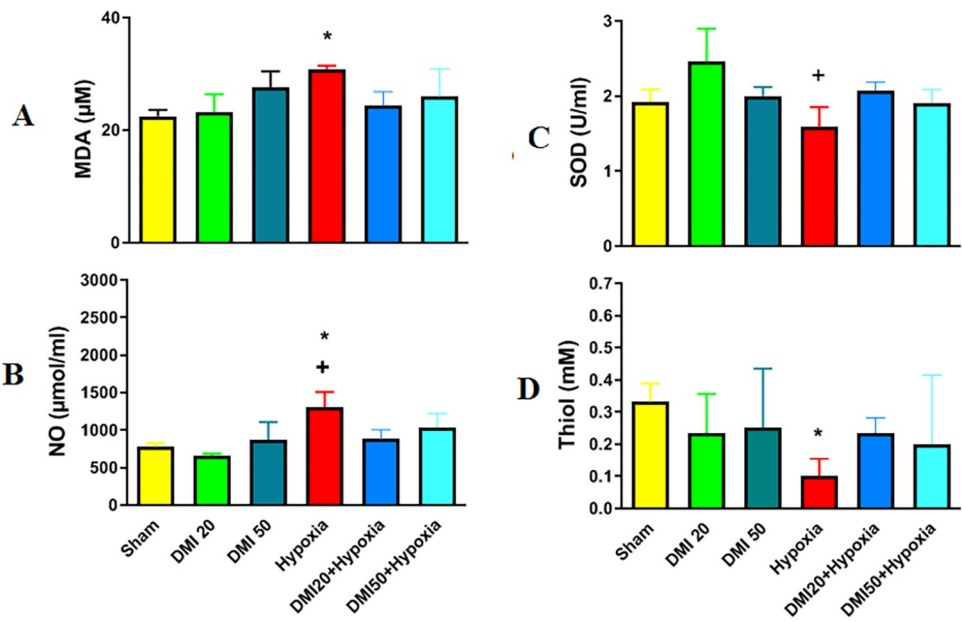

**Fig 7.** Comparison the mean of MDA (A), SOD (B), NO (C), and total Thiol concentration (D) between experimental groups. * indicates a significant difference at (p<0.05) level compared to the control group. + shows a significant difference at (p<0.05) compared to the DMI20 group. Data are presented as the mean ± SEM (n = 5).

pregnancy [45]. Therefore, the induction of oxidative stress following hypoxia is a predictable phenomenon confirmed in the present study.

Itaconate, an immunomodulatory metabolite derived from the tricarboxylic acid (TCA) cycle, increases significantly following inflammatory stimulation to limit inflammation and has been shown to reduce neuroinflammatory responses in Alzheimer's disease mouse models [27]. In this study, the levels of MDA and NO were significantly elevated in hypoxia groups compared to controls, while SOD activity was lower in the hypoxia group than in the DMI-treated group, and total thiol content was also reduced. Notably, DMI treatment normalized these oxidative stress markers, indicating its protective effect against oxidative stress in hypoxic rats.

Khadem et al. reported that treatment with DMI have the ability to enhance the GSH/GSSG ratio and improve antioxidant defense factors, such as catalase and superoxide dismutase, while reducing lipid peroxidation [46].

Electroencephalography (EEG) is a crucial technique for studying seizures in animal models of neurological disorders.

EEG evaluation was performed by comparing the power of raw EEG waves and the powers of gamma, beta, alpha, theta, and delta waves among the studied groups. In our study, no significant differences were found in gamma wave power between experimental groups. However, beta and theta wave powers were significantly decreased in the hypoxia groups compared to the control group. The hypoxic group treated with 50 mg/kg DMI showed significantly higher beta wave power than the untreated hypoxic group. Alpha wave power did not differ significantly between hypoxic and control groups. In contrast, delta wave power was significantly increased in the hypoxic group compared to controls, but it was lower in the hypoxic group receiving DMI than in the untreated hypoxic group. These results suggest that hypoxia induces notable changes in EEG wave power, and DMI treatment may help normalize these alterations.

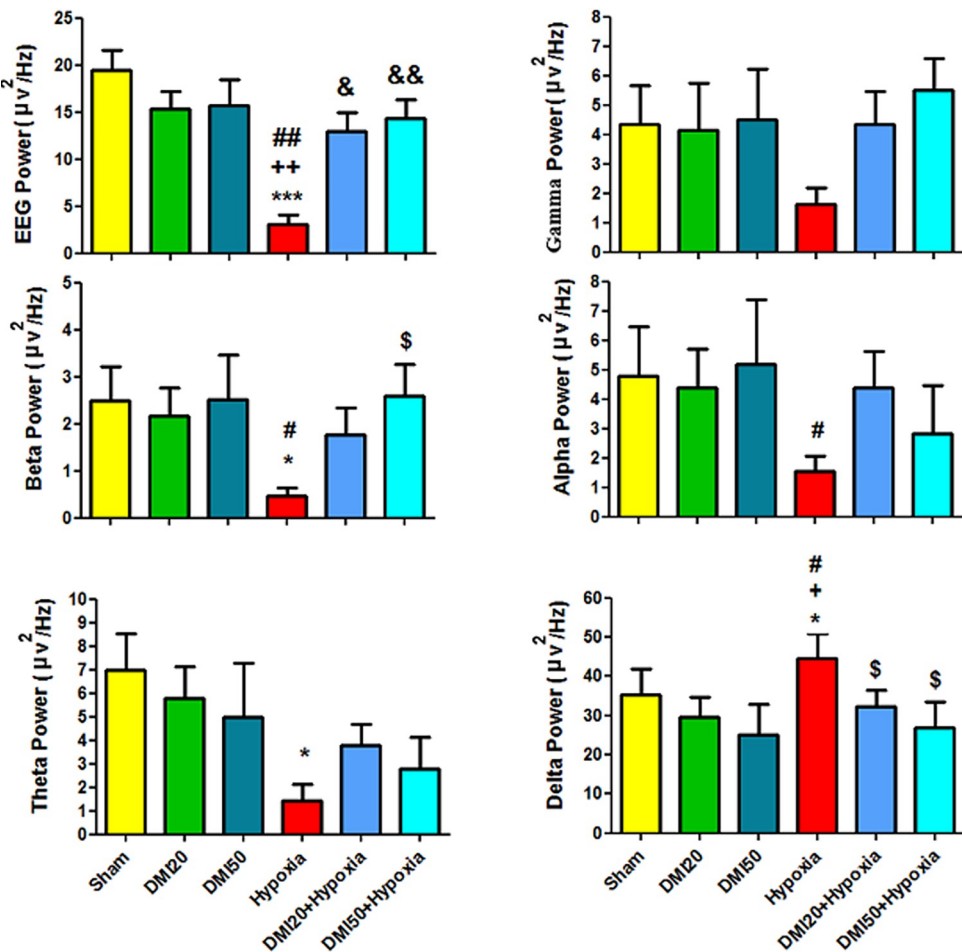

**Fig 8. Comparison the mean of EEG and its frequency bands powers (μV2/Hz).** Data present as the mean ± SEM (n = 5). A) EEG power. *** $P<0.001$ vs. sham group. ++ and ## $P<0.01$ vs.DMI20 and DMI50 groups. $ and $ $ $ $P<0.05$ and $P<0.01$ vs. hypoxia group. B) Gamma power. C) Beta power. * and # $P<0.05$ vs. sham and DMI50 groups. D) Alpha power. # $P<0.05$ vs. DMI50. E) Theta power. * $P<0.05$ vs. sham. F) Delta power. *, + and # $P<0.05$ vs. sham, DMI20 and DMI50 groups. $ $P<0.05$ vs. hypoxia group.

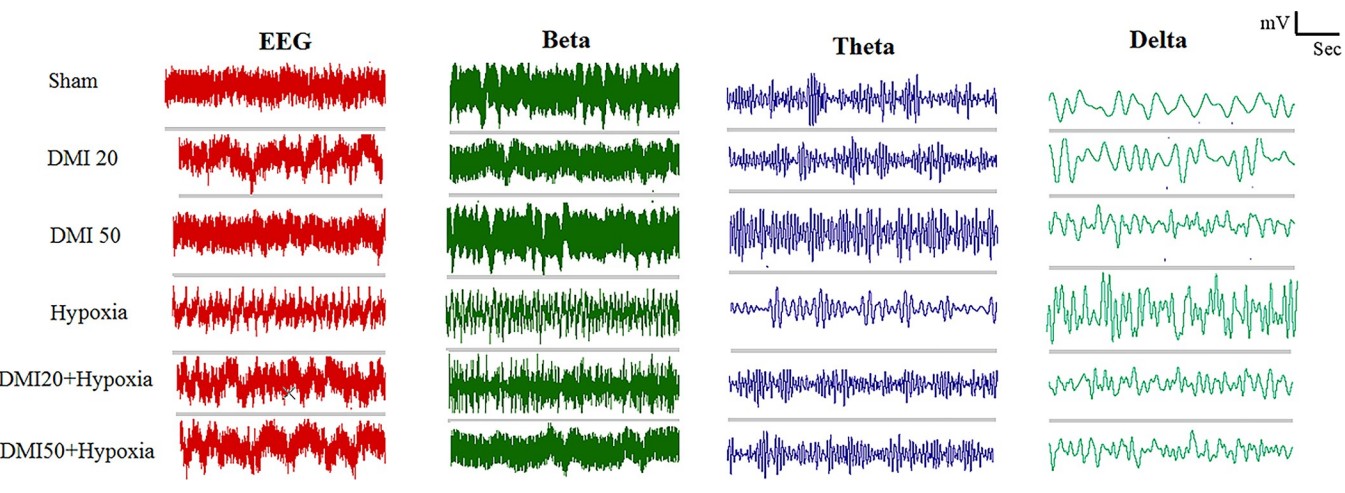

**Fig 9. The EEG power images (Crude local EEG, Beta, Theta and Delta) band traces in the experimental groups.**

In Sampath et al.'s study on rats with hypoxia-ischemia induction at P7, it was shown that the power of EEG waves decreases significantly during hypoxia and increases again over time. They demonstrated that compared to the control group, the level of EEG wave power in the hypoxic group, even on days after hypoxia induction, was significantly decreased [47]. These findings were also observed in the current study.

Budzinska and Ilasz's study in 2007 demonstrated an increase in relative EEG power in the delta frequency range and a decrease in the remaining frequency ranges following hypoxia. However, compared to the baseline level, the total EEG power gradually decreased, while the power in the delta frequency range increased in subsequent hypoxia, which is consistent with our results [48].

After hypoxia period, the decrease in total EEG power was due to the reduction in the power of theta, alpha, and beta frequencies, while the power of delta frequency slightly increased [5, 49]. Additionally, the present study demonstrated that pre-treatment with DMI can significantly contribute to the recovery of EEG wave power. No previous study had examined the effect of DMI on EEG wave power following hypoxia.

The relationship between brain wave activity (as measured by EEG) and hypoxia is complex and can vary depending on the severity and duration of the hypoxic event. In general, different types of brain waves—beta, alpha, and delta—are associated with various states of consciousness, cognitive function, and overall brain health.

In cases of hypoxia (reduced oxygen availability), the brain's electrical activity can be significantly affected by low oxygen. The decline in beta and alpha wave activity in hypoxic conditions may be attributed to impaired cognitive function and reduced alertness. Hypoxia can lead to neuronal dysfunction, impacting higher cortical functions that generate beta and alpha rhythms [50].

Research has shown that during hypoxic events, the brain prioritizes survival mechanisms over cognitive functions, leading to a suppression of higher frequency waves [51]. The increase in delta wave activity during hypoxia may reflect a shift to more primitive brain functions. Delta waves are often seen in states of unconsciousness or deep sleep, suggesting that the brain may be entering a protective mode in response to hypoxia [52]. Some studies propose that increased delta power could be a compensatory mechanism to conserve energy or maintain homeostasis under low oxygen conditions [53].

The relationship between hypoxia and changes in EEG power can be summarized as follows: Cognitive Impairment: As oxygen levels drop, cognitive functions decline, leading to decreased beta and alpha activity. Survival Mechanisms: The brain may enter a protective state characterized by increased delta activity, which is less energy-intensive. Neuronal Damage: Prolonged hypoxia can lead to neuronal injury, further altering the balance of brain wave activity.

## Conclusion

Based on the findings of the present study, treatment with DMI can be effective in improving cognitive function in neonatal hypoxic-induced seizures. It can also enhance motor coordination and balance while reducing anxiety in rodents. Furthermore, the results indicate that DMI treatment can reduce oxidative stress and inflammation in hypoxic rats and increase the power of EEG waves during adolescence. Considering the obtained results, it appears that the reduction of oxidative stress and inflammation is one of the protective mechanisms of DMI following early-life hypoxia induction. Moreover, DMI improves memory, balance, anxiety-like behavior, and EEG wave power.

## Supporting information

**S1 Raw data.**
(RAR)

**S1 Graphical abstract.**
(TIF)

## Acknowledgments

This research was extracted from Shadi Nazarizadeh's MS.c. thesis. This work was approved and carried out in the department of physiology at the Shahid Chamran University of Ahvaz and we are grateful to the Research Council of Shahid Chamran University of Ahvaz.

## Author Contributions

**Conceptualization:** Zohreh Ghotbeddin.

**Data curation:** Shadi Nazarizadeh.

**Formal analysis:** Shadi Nazarizadeh.

**Methodology:** Shadi Nazarizadeh, Samireh Ghafouri, Alireza Sarkaki.

**Software:** Alireza Sarkaki.

**Supervision:** Zohreh Ghotbeddin.

**Visualization:** Samireh Ghafouri.

**Writing – original draft:** Zohreh Ghotbeddin, Samireh Ghafouri.

**Writing – review & editing:** Zohreh Ghotbeddin, Alireza Sarkaki.

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
