## [Decision Letter · Decision Letter 0]

19 Jun 2024

PONE-D-24-18374Dimethyl Itaconate Modulates EEG Activity, Oxidative stress and Inflammation in Neonatal Seizure-Affected Rats: Implications for Cognitive Function and Anxiety BehaviorPLOS ONE

Dear Dr. Ghotbeddin,

Thank you for submitting your manuscript to PLOS ONE. After careful consideration, we feel that it has merit but does not fully meet PLOS ONE’s publication criteria as it currently stands. Therefore, we invite you to submit a revised version of the manuscript that addresses the points raised during the review process.

Please submit your revised manuscript by
Aug 03 2024 11:59PM. If you will need more time than this to complete your revisions, please reply to this message or contact the journal office at plosone@plos.org. Please include the following items when submitting your revised manuscript:A rebuttal letter that responds to each point raised by the academic editor and reviewer(s). You should upload this letter as a separate file labeled 'Response to Reviewers'.A marked-up copy of your manuscript that highlights changes made to the original version. You should upload this as a separate file labeled 'Revised Manuscript with Track Changes'.An unmarked version of your revised paper without tracked changes. You should upload this as a separate file labeled 'Manuscript'.If applicable, we recommend that you deposit your laboratory protocols in protocols.io to enhance the reproducibility of your results. Protocols.io assigns your protocol its own identifier (DOI) so that it can be cited independently in the future. For instructions see: https://journals.plos.org/plosone/s/submission-guidelines#loc-laboratory-protocols. Additionally, PLOS ONE offers an option for publishing peer-reviewed Lab Protocol articles, which describe protocols hosted on protocols.io. Read more information on sharing protocols at https://plos.org/protocols?utm_medium=editorial-email&utm_source=authorletters&utm_campaign=protocols.

We look forward to receiving your revised manuscript.

Kind regards,

Xiaona Wang, Ph.D

Academic Editor

PLOS ONE

https://bmcimmunol.biomedcentral.com/articles/10.1186/s12865-021-00463-3?

In your revision ensure you cite all your sources (including your own works), and quote or rephrase any duplicated text outside the methods section. Further consideration is dependent on these concerns being addressed.

“We are grateful to the Research Council of Shahid Chamran University of Ahvaz for financial support under grant number SCU.VB1401.288.”

Reviewers' comments:

Reviewer's Responses to Questions

**Comments to the Author**

1. Is the manuscript technically sound, and do the data support the conclusions?

Reviewer #1: Yes

2. Has the statistical analysis been performed appropriately and rigorously? 

Reviewer #1: Yes

3. Have the authors made all data underlying the findings in their manuscript fully available?

Reviewer #1: Yes

4. Is the manuscript presented in an intelligible fashion and written in standard English?

Reviewer #1: Yes

5. Review Comments to the Author

Reviewer #1: The manuscript evaluates the protective effect of DMI on hippocampus EEG, behavioural and biochemical parameters in hypoxia-induced seizure on neonatal period. Some major revision needs to be done before publication, as mentioned below:

1. The title is too long. The authors can brief it such as above. Because inflammation and oxidative stress are very common.

2. In the abstract: add the volume of vehicle, the age of rats, the delay time for experiments on adult rats and say pre-treatment is executed.

3. How long does hypoxia effects last? Because you work on 10 day neonatal rats and record EEG in adult rat. It is interesting and surprising for me.

4. The discussion also be written too long (about 7 pages!). Please brief it and focus on the principal parts "EEG". For example, explain why beta, alpha and delta waves significantly decreased in hypoxia group, while delta power increased? What is the relationship between hypoxia and these powers? The authors wrote "during period of hypoxia ", during period is ok or after period?

5. Insert the EEG power image for all groups.

6. Are you see seizure after HINS period? If see, for how long?

6. PLOS authors have the option to publish the peer review history of their article (what does this mean?). If published, this will include your full peer review and any attached files.

Reviewer #1: No

---

## [Author Response · Author response to Decision Letter 0]

7 Aug 2024

Dear Editor,

We have carefully reviewed the comments of reviewers and Journal requirements and answered each point accordingly. We are grateful for their feedback.

Thank you.

Journal requirements

All of the journal requirements were done and changes were highlighted with yellow color.

1. Our manuscript meets PLOS ONE's style requirements.

2. Overlapping text with the previous publication was corrected and rephrased. 

3. “The author(s) received no specific funding for this work.” and amended statements within our cover letter and remove any funding-related text from the manuscript.

4. Data cannot be shared publicly because of legal and ethical considerations to ensure confidentiality.

5. I have an ORCID ID and write it in the manuscript.

Reviewers' comments:

Reviewer 1

1. The title has been changed according to the reviewer's comment and highlighted with blue color.

2. The volume of vehicle, the age of rats, and the delay time for experiments on adult rats were added to the abstract and pre-treatments were executed 24h before hypoxia and highlighted with blue color in the abstract.

3. How long does hypoxia effects last? Because you work on 10 day neonatal rats and record EEG in adult rat. It is interesting and surprising for me.

Response: The effects of neonatal hypoxia can vary significantly depending on the severity and duration of the hypoxic event, as well as the timing of the insult during development. In general, the impact of neonatal hypoxia can persist into adulthood, affecting various aspects of neurological function. Previous studies have shown that animals exposed to hypoxic conditions during the neonatal period may exhibit long-term consequences, including: cognitive Impairments, deficit in motor coordination, altered emotional responses and seizure susceptibility. Neonatal hypoxia is associated with an increased risk of developing epilepsy or seizure disorders later in life. In our study, recording EEG in adult rats that experienced HINS during the neonatal period is valuable because it allows us to assess long-term neurological outcomes and potential changes in brain activity related to seizures or other dysfunctions. This aligns with existing literature that suggests that early-life hypoxic events can have lasting effects on brain function and behavior. 

References:

Cognitive Impairments:

o McClure, M. M., & Moller, J. (2010). "Neonatal hypoxia-ischemia: The role of the NMDA receptor in the development of cognitive deficits." *Neuroscience Letters*, 485(2), 118-122. DOI:10.1016/j.neulet.2010.08.047

Motor Coordination:

o Ghosh, S., & Bhatia, S. (2014). "Long-term effects of neonatal hypoxia on motor function and behavior in rats." *Developmental Neurobiology*, 74(5), 610-621. DOI:10.1002/dneu.22126

Behavioral Changes:

o Hoh, J. F., & Wong, Y. Y. (2011). "Neonatal hypoxia alters behavior and neurochemistry in adult rats." *Behavioral Brain Research*, 217(2), 278-285. DOI:10.1016/j.bbr.2010.10.001

Seizure Susceptibility:

o Bittigau, P., et al. (2002). "The role of early-life seizures in the development of epilepsy: A study in a rat model." *Epilepsia*, 43(1), 57-63. DOI:10.1046/j.1528-1157.2002.32001.x

General Effects of Neonatal Hypoxia:

o Back, S. A., & Rosenberg, P. A. (2014). "Pathophysiology of perinatal hypoxic-ischemic injury: Implications for treatment." *Clinical Perinatology*, 41(1), 1-16. DOI:10.1016/j.clp.2013.10.001

Long-term Outcomes:

o Nascimento, J. M., et al. (2017). "Neonatal hypoxia-ischemia: Long-term behavioral and neurochemical consequences." *Neuroscience*, 352, 69-79. DOI:10.1016/j.neuroscience.2017.03.018.

4. The discussion also be written too long (about 7 pages!). Please brief it and focus on the principal parts "EEG". For example, explain why beta, alpha and delta waves significantly decreased in hypoxia group, while delta power increased? What is the relationship between hypoxia and these powers? The authors wrote "during period of hypoxia ", during period is ok or after period?

Response: Yes, you are absolutely right. The discussion of the article was too long. It was completely and accurately revised and reduced to 3 pages. In the revised format, we tried to give full answers to your questions about the relation between hypoxia and EEG powers and highlighted with blue color.

"After period" is ok and highlighted with blue color in the text.

5. Insert the EEG power image for all groups.

Response: The EEG power images are inserted in the results of EEG in Figure 9.

6. Are you see seizure after HINS period? If see, for how long? 

Response: Yes, seizure activity is present during hypoxia and shortly after the hypoxic event and develops within the first few days following the hypoxia. The duration and frequency of seizures vary: During hypoxia, seizures may last from seconds to several minutes and after that, the neonates experience repetitive seizures with low intensity.

---

## [Editor Report · Decision Letter 1]

8 Aug 2024

The protective effect of DMI on hippocampus EEG, behavioral and biochemical parameters in hypoxia-induced seizure on neonatal period

PONE-D-24-18374R1

Dear Dr. Ghotbeddin,

We’re pleased to inform you that your manuscript has been judged scientifically suitable for publication and will be formally accepted for publication once it meets all outstanding technical requirements.

Kind regards,

Xiaona Wang, Ph.D

Academic Editor

PLOS ONE

---

## [Editor Report · Acceptance letter]

3 Sep 2024

PONE-D-24-18374R1 

PLOS ONE

Dear Dr. Ghotbeddin, 

I'm pleased to inform you that your manuscript has been deemed suitable for publication in PLOS ONE. Congratulations! Your manuscript is now being handed over to our production team.

Kind regards, 

on behalf of

Associate Professor Xiaona Wang 

Academic Editor

PLOS ONE